# Deciphering the Metabolic Basis and Molecular Circuitry of the Warburg Paradox in Lymphoma

**DOI:** 10.3390/cancers16213606

**Published:** 2024-10-25

**Authors:** Dashnamoorthy Ravi, Athena Kritharis, Andrew M. Evens

**Affiliations:** 1Rutgers Cancer Institute of New Jersey, Rutgers University, New Brunswick, NJ 08901, USA; 2ICON Medical Affairs, Blue Bell, PA 19422, USA; athena.kritharis@iconplc.com

**Keywords:** Warburg effect, lymphoma, lactate, glucose, nucleotides, glutamine, targeted inhibitors, transaminase

## Abstract

This study explores Warburg’s paradox, whereby cancer cells use both glucose and oxygen to survive, even though glucose is converted to lactate instead of being oxidized. By systematically investigating cellular metabolism during each phase of cell division and comparing the metabolic profiles of lymphoma cells and non-malignant lymphocytes, we discovered that pyruvate, the end-product of glucose metabolism, is converted into alanine. This conversion directs glutamine carbon, rather than glucose, into the TCA cycle. Furthermore, using fludarabine to selectively inhibit lymphoma cell proliferation, we showed that blocking the conversion of pyruvate to alanine disrupts the TCA cycle and interferes with the supply of nucleotides and the energy necessary for cancer cell growth. Our findings suggest that with the suppression of glucose oxidation, the conversion of pyruvate to alanine is the crucial metabolic link that connects glucose and oxygen metabolism and serves as a key component of Warburg’s paradox.

## 1. Introduction

“At first glance, it appears paradoxical that a cell that can live by fermentation may die because it lacks oxygen”. “But there is really no contradiction here” and “Providing glucose to the tumor is bad, but providing oxygen is even worse” [1]. Otto Warburg made these proverbial statements exactly a century ago (in 1923), following his seminal discovery that tumors secrete lactic acid when aerobic glycolysis is present [1]. Warburg subsequently theorized that metabolic failure associated with respiratory failure is preceded by an increase in lactate synthesis (through fermentation) [2], but this concept was later disproven. While the existence of the “Warburg effect” has been acknowledged in numerous studies, contradictory opinions and disagreements also exist in the literature [3,4,5,6]. Nonetheless, metabolic dysfunction is recognized as an important hallmark of cancer. The importance of the Warburg phenomenon is evidenced by the integration of this concept into the clinical application of ^18^F-fluorodeoxyglucose PET (positron emission tomography) imaging (FDG-PET) [3], the adaptation of lactate and lactate dehydrogenase (LDH) in cancer diagnosis [7,8], and the development of metabolism-targeted therapeutics [9].

Several attempts aimed at resolving the mechanistic basis of the Warburg effect were predominantly focused on ATP generation, genetic, oncogenic upstream regulators, biomass, pH, signaling changes, tumor microenvironment, and interorgan metabolic interactions [3,4,9]. Most cancers exhibit increased aerobic glycolysis and lactate synthesis; however, these biochemical characteristics are also observed in many proliferative mammalian cells, yeasts, and bacteria. An increase in lactate synthesis paralleled with S phase in proliferative *Saccharomyces cerevisiae*, as well as increased glycolysis and DNA synthesis in mouse fibroblasts have been observed [10]. Although aerobic glucose to lactate conversion is not as efficient as oxidative phosphorylation for energy production, combined with the high levels of LDH protein expressed, it still has been considered a significant source of ATP production in cancer [11]. It can be argued that uncoupling glycolysis from the tricarboxylic acid cycle (TCA cycle) is crucial for providing the glycolytic intermediates necessary for the biosynthesis of lipids, amino acids, and nucleotides in malignant cells [6]. Although the conversion of pyruvate to lactate limits further oxidation of glucose and results in the uncoupling of glycolysis from the TCA cycle, the TCA cycle and oxidative phosphorylation remain active in cancer cells [12,13]. In addition, the TCA cycle can be sustained by anaplerosis to maintain the electron transport chain and oxidative phosphorylation [14]. It is possible for cancer cells to use both LDH and oxidative TCA cycle mechanisms simultaneously to meet their higher energy requirements.

Recent research by Luengo et al. highlights the significance of cancer cells converting glucose into lactate even when oxygen is available [15]. In this study, aerobic glycolysis is found to facilitate the balancing of redox reactions by producing NADPH and regenerating NAD+, which facilitates the synthesis of metabolic precursors to enhance cell proliferation [15]. Consequently, aerobic glycolysis supporting energy metabolism is considered less important than biosynthesis [15]. Furthermore, other studies have shown that the glucose uptake, which facilitates energy production through lactate dehydrogenase (LDH) and maintains redox balance, also supports the oxidative branch of the pentose phosphate pathway [16]. While ATP is normally associated with energy supply, increased ATP production has been shown to negatively impact proliferation in PTEN-deficient mouse embryonic fibroblasts and prostate cancer cells [17,18]. Conversely, blocking ATP synthesis is well-established as an effective method of inhibiting the proliferation of cancer cells [19]. As a result, it is unclear whether the Warburg metabolism is primarily responsible for affecting energy metabolism and/or biosynthesis [20].

During normal cell proliferation, energy and biosynthesis are tightly regulated and proceed in an orderly manner [20]. To proliferate rapidly, malignant cells, however, require energy and biosynthesis simultaneously [20]. The conversion of pyruvate to lactate is therefore mechanistically preferable to its oxidation by cancer cells [21]. In this manner, the TCA cycle intermediates, and excess ATP is prevented from leading to feedback inhibition, ensuring continuous glucose uptake and glycolytic activity (see graphical abstract). Based on these contexts, we focused our efforts on understanding the mechanistic basis for Warburg’s paradox in this study.

Warburg’s paradox influences malignancy aggressiveness; it is important to understand the metabolic connection between glucose utilization, glucose oxidation, and oxygen metabolism. Moreover, TCA cycle intermediates such as acetyl CoA, α-ketoglutarate, citrate, and fumarate are implicated in tumorigenic activities [22]. This could lead to the continuous loss of metabolic intermediates, which in turn raises questions regarding whether the fidelity of the TCA cycle is maintained during malignancy. Most importantly, such events could impair the ability of the TCA cycle to provide oxaloacetate for accepting carbons (from acetyl-CoA) for glucose oxidation and energy production. Therefore, understanding how cancer cells can disconnect from glucose and oxygen metabolism while remaining dependent on them becomes more challenging. Cancer cells undergo extensive metabolic reprogramming to circumvent rate-limiting mechanisms for uninterrupted glucose uptake [23], but these mechanisms are necessary to simultaneously support anabolic functions and energy production. There is still a lack of precise understanding of these reprogrammed metabolic pathways within the context of the Warburg paradox, which is crucial for the development of more effective cancer treatments.

Our investigation into this aspect originated from a perspective obtained from our prior study, which revealed that citrate exiting the TCA cycle is utilized by fatty acid synthase (FASN) for de novo lipogenesis (DNL) [24]. By enhancing the turnover rate of NADPH/NADH, this metabolic activity enables nucleotide synthesis via the pentose phosphate pathway (PPP) [24]. The results of this study prompted us to investigate metabolic compensatory responses for citrate loss, which may be responsible for the reduced glucose oxidation and increased lactate synthesis. Further, the metabolic function, which compensates for citrate loss, may be responsible for the continuity of oxygen-dependent energy metabolism (via the TCA cycle). Together these mechanisms may provide the biological basis of Warburg’s paradox. Therefore, in this study, our goal is to understand the metabolic relationship between glucose and nucleotide biosynthesis in the context of citrate loss and TCA cycle functionality under these conditions. Our approach to resolve these metabolic relationships involved two complementary approaches. The first was to identify the metabolic signatures that increase with cell proliferation across the cell cycle phases and determine whether they differ between malignant cells (lymphoma cell lines and tumors) and normal cells (represented as non-malignant lymphoblastoid (LCL) and normal tissues). By using bioinformatically selected targets, the second approach identifies metabolic functions that become downregulated while blocking proliferation selectively in the malignant lymphoma cells, but not in the non-malignant LCL. In these experiments, we used whole cell extracts for mass spectrometry to estimate the size of the total pool of polar metabolites and investigated the relative carbon contributions from ^13^C_1,2_ D-Glucose and ^13^C_5_,^15^N_2_ L-Glutamine. The isotope ^13^C_1,2_ D-glucose is commonly used to track carbon flow into nucleotides (from C_2_-glucose to C_1_-ribose, with a loss of a single carbon) through the oxidative pentose phosphate pathway (PPP), while C_2_-glucose remains C_2_ throughout glycolysis, and it gradually loses its carbon through the TCA cycle, as described in the literature [25,26,27]. The use of ^13^C_5_,^15^N_2_ L-Glutamine was utilized in order to detect N_1_ nitrogen transfers to alanine or aspartate during transamination [28], as well as to determine the relative contribution of glucose and glutamine carbons in various metabolic pathways. Through identification and comparison of metabolic profiles associated with proliferation and malignancy, but suppressed by proliferation inhibition, we aim to establish a metabolic link between aerobic glycolysis and the TCA cycle.

## 2. Materials and Methods

### 2.1. Cell Culture and Reagents

ATCC (STR profiling) authenticated lymphoma cell lines, CA46 and SUDHL4, and transformed human primary B lymphoblastoid cell line (LCL), purchased from Astarte Biologics, were grown in the RPMI 1640 medium with 10% heat-inactivated fetal bovine serum (FBS) and 200 U of penicillin/streptomycin (Mediatech, Manassas, VA, USA) under 5% CO_2_ and at 37 °C. Metabolic pathway inhibitors Abexinostat (#S1090), Fludarabine (#S1491), Thioguanine (#S1774), YK-4-279 (#S7679), Auranofin (#S4307), Sodium oxamate (#S6871), CPT1-2 (#S2968), MK801 (#S2857), and XAV939 (#S1180) were purchased from Selleckchem (Houston, TX, USA). Demvistat (#HY-15453) and MCT1-III (#5387590001) were purchased from Medchem Express (Manmouth Junction, NJ, USA) and Calbiochem (San Diego, CA, USA), respectively. RPMI-1640 glucose and pyruvate deficient (Catalog# R1383), RPMI-1640 glutamine deficient (Catalog# R5886), and dialyzed fetal bovine serum (Catalog#F0392), were purchased from Sigma (St. Louis, MO, USA). ^13^C_1,2_, D-Glucose (CLM-504-PK) and ^13^C_5_,^15^N_2_, L-Glutamine (CNLM-1275-H-PK) were purchased from Cambridge Isotopes, (Tewksbury, MA, USA).

### 2.2. Flow Sorting of Cell Cycle Phases

A total of 15 × 10^6^ CA46 were cultured without changing growth medium for 48 h in T225 flask, followed by labeling with 2 μg/mL Hoechest 33342 (Life Technologies, Eugene, OR, USA) for 45 min, by directly adding to the existing medium. Then, cells were centrifuged at 800 rpm for 5 min and the medium was replaced with chilled PBS consisting of 5 × 10^6^ cells/mL and maintained on ice until completion of flow sorting. Using a total of ~50 × 10^6^ cells, flow sorting of “singlet” cells based on Hoechest 33342 stain gated as G1, S and G2 were sorted into separate tubes and maintained on ice, until flow sorting was completed. Then, the flow sorted cells were centrifuged and resuspended at 0.5 × 10^6^ per well in 6 well plates, plated in triplicates, acclimatized for one hour at 37 °C and 5% CO_2_, followed by incubation for 2 h in the pyruvate deficient RPMI medium consisting of dialyzed serum for pool size estimation. Through comparing the metabolic profiles of the cell cycle phase sorted with unsorted flow, through cells and cells directly collected from cell culture, we ensured that the flow sorting procedure did not introduce any metabolic artifacts in these experiments. To avoid toxicity and to prevent confounding effects on metabolic profiles, propidium iodide was not added to exclude dead cells. For isotope tracer investigations with flow, sorted cells were incubated for 2 h or for other experiments and inhibitor treatments; cells were incubated for 12 h in medium consisting of the following isotope tracers, ^13^C_1,2_-glucose or ^13^C_5_, ^15^N_2_-glutamine, consisting of the appropriate metabolite deficient RPMI-1640 medium, supplemented with 10% dialyzed fetal bovine serum. Since flow-sorted cells could progress to the next phase in cell culture, metabolic labeling and pool size analysis were limited to two-hour incubations to ensure metabolic estimates remained consistent across each phase.

### 2.3. Preparation of Tumor Specimens for Metabolomic Profiling

Cryopreserved diffuse large B cell lymphoma (DLBCL) tumor and normal lymph nodes were obtained as deidentified specimens through Biospecimen Repository and Histopathology Service, following the Scientific Review Board of Rutgers Cancer Institute of New Jersey’s institutional review board exempt approval for study protocol #001835 on 14 December 2019. Lymphoma tumor and normal lymph node tissues were sliced using sterile scalpel blades on Petri dishes placed on a dry ice tray. Tissue bits of <25 mg were weighed and powdered using Cryomill and sterile zirconium beads as two sets. Powdered tissues were extracted for polar metabolite analysis using an appropriate volume of extraction buffer, i.e., methanol:acetonitrile:water 40:40:20 consisting 0.5% formic acid. Following 5 min of incubation on ice, the tissue was neutralized with 15% ammonium bicarbonate and then centrifuged at 14,000 rpm for 15 min at 4 °C. Supernatants were collected, flash-frozen, and stored in −80 °C until mass spectrometry. The second set of tissue powder was extracted with cell lysis buffer (Cell lysis buffer, Cell signaling Technology, Beverly, MA, USA). Protein content was normalized by tissue weight, determined using Biorad protein assay reagent (Hercules, CA, USA), and then utilized for normalizing the tissue weight of samples used in the mass spec analysis. Determination of accuracy in peak detection, quantification, and validations with reference metabolites were performed, as reported previously [24,29].

### 2.4. Metabolomic Profiling

Whole cell lysates were prepared from cultured cells using the same extraction method as for tumors (above) and as previously reported [24]. Whole cell lysates were used in metabolomic profiling experiments, since the focus of this study is to investigate intracellular metabolic pathways. To compare relative changes for the same metabolite across different experimental conditions, we used estimated pool sizes (corrected for natural abundance and normalized ion counts) and fractional labeling indices. Estimation of pool sizes without labeling were performed using lymphoma cell lines or LCL cells cultured in RPMI-1640 (Mediatech, Manassas, VA, USA) and 10% dialyzed fetal bovine serum (Catalog# F0392, Sigma Aldrich, St. Louis, MO, USA). For the isotope tracer investigations, parallel labeling experiments using RPMI-1640 glucose and pyruvate deficient (Catalog# R1383), substituted with 2 g/L ^13^C_1,2_, D-Glucose, or RPMI-1640 glutamine deficient (Catalog# R5886), substituted with 0.3 g/L ^13^C_5_, ^15^N_2_, L-Glutamine and dialyzed fetal bovine serum (Catalog# F0392) Sigma (St. Louis, MO, USA), were performed. ^13^C_1,2_-glucose was utilized for comparing ^13^C enrichments to distinguish ^13^C labeled C1 isotopomers derived from the oxidative pentose phosphate pathway (PPP) [25,26,27]. Additionally, ^13^C_1,2_-glucose allows for distinguishing TCA cycle intermediates, obtained directly from pyruvate ^13^C_2_ in citrate or ^13^C_1_ in malate generated in the TCA cycle, through loss of carbon from each round of this metabolism. ^13^C_5_, ^15^N_2_-Glutamine was utilized for comparing ^13^C enrichments in the TCA cycle and carbon transfers, if any into glycolysis, and identify the ^15^N products yielded from the transamination reaction, as described in the literature [28]. Isotope tracer experiments with ^13^C_1,2_-Glucose and ^13^C_5_, ^15^N_2_-Glutamine were performed by incubation for 12 h, so that incorporation of these labels in the nucleotides could be detected, which occurs relatively slowly, as previously observed [24]. Relative contribution of carbons were calculated by dividing the metabolite-specific carbon enrichment by the sum of corresponding carbon enrichments of relevant metabolites or different carbon sources, as described in the literature [27]. Briefly, samples extracted with (40:40:10) methanol, acetonitrile, and water, consisting of 0.5% formic acid and neutralized with sodium bicarbonate, were used for the analysis by mass spectrometry. LC−MS was performed using the Q Exactive PLUS hybrid quadrupole-orbitrap mass spectrometer (Thermo Scientific, Waltham, MA, USA), coupled to hydrophilic interaction chromatography; metabolite feature extraction using MAVEN for isotope labeled metabolites; accuracy, abundance, and correction for impurity were performed using AccuCor written in R as described [24]. The corrected ion counts were normalized by cell number or protein concentration as described before [24].

### 2.5. Western Blot

Preparation of protein lysates and Western blots were performed as described before [24], using the following primary antibodies against LDHA (RRID:AB_2066887), LDHB (RRID:AB_1124720), PDH (RRID:AB_2162928), GPT1/ALT RRID:AB_10710382), GPT2/ALT2 (RRID:AB_2927429), JUN (RRID:AB_10949318), STAT1(RRID:AB_2737027), HDAC1 (RRID:AB_2756821), NONO (RRID:AB_2940779), DRAP1 (RRID:AB_2940780), ETS1, (RRID:AB_831289), and β-actin (RRID:AB_330288), purchased from Cell Signaling Technology (Beverly, MA, USA) or Santa Cruz Biotechnology (Santa Cruz, CA, USA).

### 2.6. Cell Proliferation Assays

MTT assays were performed using DLBCL cells treated with the appropriate pharmacological inhibitors for 72 h, as described before [24], using CellTitre 96, non-radioactive cell proliferation kit purchased from Promega (Madison, WI, USA). Briefly, 10^4^ cells/100 µL were plated in a 96-well plate and treated with increasing concentrations of appropriate drugs and incubated at 37 °C and 5% CO_2_ for 72 h. MTT assay was then performed following the instructions supplied by the manufacturer, and the absorbance was measured at 570 nM, using SpectraMax M5 plate reader (Molecular Devices, San Jose, CA, USA). IC_50_ values for drug treatments were derived using GraphPad Prism (Boston, MA, USA).

### 2.7. Transcriptomics and Bioinformatic Analysis

RNA isolation and transcriptomic datasets for lymphoma cell lines, used in mapping the Warburg effect on molecular-metabolic circuitry, were obtained from our previous research [24,30,31,32]. All transcriptomic assessments were performed in biological triplicates using Affymetrix Human Genechip 2.0 ST or Human HT 12 Genechip Illumina. These datasets included the following DLBCL cell lines: Raji, SUDHL2, SUDHL4, SUDHL6, SUDHL10, OCI-LY3, Jurkat, L540, L428, and Hut78 cell lines. The raw data for these experiments are available at the NCBI Gene Expression Omnibus database, with the following identifiers: GSE102760 GSE102764, GSE66417, GSE66415, and GSE126768 (see Appendix A), and the normalized data is included in Appendix A. Meta-analyses of lymphoma patient tumors were performed using harmonized median normalized log_2_ transformed transcriptomic datasets, retrieved from the National Cancer Institute (NCI) dataset consisting of DLBCL (*n* = 574), accessible from portal.gdc.cancer.gov, as reported in this study [33]. Upstream regulatory factors for metabolic genes were retrieved from the Genecard-linked GeneHancer database as a list of high-confidence enhancers and promoters associated with every metabolic gene published on http://www.genecards.org, accessed on 16 September 2020. Network analysis of metabolic gene-regulatory factors was performed using Cytoscape version 3.8.2, as described before [24].

### 2.8. Statistical Analysis

All experiments were performed in triplicate. The metabolic pool sizes were estimated from three independent experiments analyzed in triplicate. Significant differences (by log_2_-fold change) between control and treatment were statistically determined by one-way analysis of variation (ANOVA) and post hoc analysis, using log_2_ transformed pool size intensities from data normalization performed using the software packages included in Metaboanalyst 3.0 [34]. Identification of top significant metabolite features was performed by partial least squares—discriminant analysis (PLS-DA) and variable importance in projection (VIP) scoring analysis, following principal component analysis (PCA). Statistical correlation analysis and Spearman rank correlation were performed using the software packages included in Metaboanalyst 3.0 [34]. The statistical analysis for transcriptomic datasets was performed as previously described [30,31]. Heatmap analysis represented as row variances, hierarchical clustering based on one minus Pearson’s correlation by predefined groups, and heatmap collapsing based on pathway metabolite median were performed using Morpheus, available from https://software.broadinstitute.org/morpheus/, accessed on 16 September 2020.

## 3. Results

### 3.1. Levels of Lactate and Nucleotides Surge at S Phase

In malignancy, glucose is converted to lactate at a higher rate, and Warburg also noted that both glucose and oxygen are essential for cancer survival [1]. Since malignancy involves uncontrolled proliferation, we first investigated the metabolic dynamics associated with lymphoma cell cycle progression. We therefore performed metabolomic profiling of flow sorted log phase Hoechst 33342-labeled CA46 according to the G1, S, and G2 phases as shown in Figure 1a. The results of metabolomic profiling of CA46 by cell cycle phases consisting of metabolic pool sizes, represented as heatmaps clustered by Spearman rank correlation analysis, revealed that most amino acids and glucose pools are relatively larger during the G1 and S phases. Based on one-way ANOVA, false discovery rate (FDR 0.05) and Fisher’s least significant difference (LSD) analysis, 31 metabolites representing glycolysis, TCA cycle intermediates, and nucleotides were identified as significantly increased in the S phase along with lactate, and nucleotides remained elevated through G2 (Figure 1b and Appendix A). Additionally, amino acid pools decreased from the G1 to S and G2 phases of the cell cycle (Figure 1b and Appendix A). Interestingly, among the amino acids, we observed that alanine, glutamate, aspartate, and proline metabolic pools were higher in the S and G2 than in the G1 phase (Figure 1b).

Further, a VIP score analysis (see “Methods”) of the overall metabolic changes vs. the cell cycle progression also identified α-ketoglutarate, lactate, and alanine as among the top 10 metabolites associated with increases in nucleotide pool sizes (UMP, GMP, ATP, CTP, and CMP) at the S phase (Figure 1c). Since alanine, aspartate, α-ketoglutarate, glutamate, and pyruvate were increased during the S phase (Figure 1b,c), we hypothesized that glutamine/glutamate carbon entry as α-ketoglutarate could be facilitated by transaminases (alanine and aspartate transaminases), allowing glucose to indirectly support the TCA cycle. Therefore, our next objective is to determine whether these transaminases mediated metabolic functions are active during the S phase.

### 3.2. Pyruvate and Alanine Transaminase

To determine whether pyruvate–alanine transamination provides glutamine carbons to sustain the TCA cycle during S phase, we sorted cells based on their cell cycle phase and performed isotope enrichment analysis using ^13^C_1,2_-Glucose and ^13^C_5,_^15^N_2_-Glutamine, as illustrated in Figure 2a. The glycolytic metabolism of glucose with ^13^C_1,2_ yielded two molecules of pyruvate, with one molecule enriched with ^13^C_1,2_ and the other remaining unlabeled. Alanine and lactate retained the same isotopic enrichments since there was no carbon loss during the metabolic conversion from pyruvate, as shown in Figure 2a. Upon entering the TCA cycle, and accruing loss of one carbon as CO_2_, α-ketoglutarate was enriched with ^13^C_1_, and this enrichment could then transfer to glutamate by pyruvate-dependent alanine transaminase. However, oxaloacetate remained unlabeled due to the loss of another carbon through the further metabolism of α-ketoglutarate to yield oxaloacetate in the TCA cycle, as summarized in Figure 2a.

In parallel labeling experiments with ^13^C_5_,^15^N_2_-Glutamine, loss of one nitrogen yielded ^13^C_5_,^15^N_1_-glutamate (through glutaminase), which would then transfer ^15^N_1_ enriching alanine (C_0_^15^N_1_ or C_3_^15^N_1_) or aspartate (C_0_^15^N_1_ or C_4_^15^N_1_) from glucose- or glutamine-derived carbon backbones by participating in transaminase reactions (Figure 2a). Additionally, ^13^C_5_ α-ketoglutarate (derived from glutamate) participating in the TCA cycle and incurring carbon loss converted into ^13^C_4_ oxaloacetate (Figure 2a). Furthermore, this ^13^C_4_ labeled oxaloacetate could produce pyruvate through malic enzyme, which could result in enriching both lactate and alanine with ^13^C_3_ (Figure 2a).

Following 2 h of culture in ^13^C_1,2_ Glucose medium without pyruvate, the cell cycle sorted CA46 cells showed a significant increase in pyruvate, lactate, alanine, and α-ketoglutarate pool sizes from G1 to S (*p* < 0.05), and then decreased by G2 (Figure 2a). Under these conditions, glutamate and aspartate pool sizes showed modest changes. Pyruvate, lactate, and alanine were all enriched with ^13^C labels from glucose-derived carbons (Figure 2a). However, aspartate and α-ketoglutarate did not contain these ^13^C labels, and prior studies have demonstrated that the labeling of aspartate occurs at slower kinetics [35]. Cell cycle sorted CA46 cells labeled with ^13^C_5_,^15^N_2_-Glutamine showed significant increases in glutamate and α-ketoglutarate levels during the S phase compared with the G1 phase (*p* < 0.05) (Figure 2a). Interestingly, we observed that only one-third of glutamate carbons were enriched with ^15^N, represented as ^13^C_5_N_1_, while the remaining two-thirds contained ^13^C_5_N_0_, indicating that glutamate carbons undergo rapid turnover, incurring ^15^N loss (Figure 2a). Moreover, 78% of α-ketoglutarate was enriched with ^13^C_5_, derived from glutamine, whereas glucose-derived ^13^C carbons remained undetected, suggesting glutamine is the main carbon source for α-ketoglutarate in TCA during the S phase of the cell cycle (Figure 2a).

Next, we observed that with increases in both alanine and aspartate levels, an increase in transamination activity resulted in a 1.5-fold increase in enrichment with ^15^N to C_0_ alanine (from glucose-derived) in the S phase compared to G1 (*p* < 0.05) (Figure 2a). In contrast, by averaging all three phases of the cell cycle, aspartate contained 20–30% carbon obtained from glutamine, of which only 10% were labeled with ^15^N, without any increases at the S phase compared to G1 (Figure 2a). Overall, since ^13^C enrichments from glucose were absent in α-ketoglutarate, and a significant 3-fold increase in α-ketoglutarate consisting of carbon enriched from glutamine were observed in the S phase, along with ^15^N enrichment increases detected in alanine, we conclude glucose-dependent alanine transaminase facilitates participation of glutamine carbon into the TCA cycle, through the S phase transition during cell cycle. Considering that ^13^C labeling from glucose was not detected in α-ketoglutarate, oxaloacetate, and aspartate, and ^13^C carbons derived from glutamine were not detected in pyruvate, lactate, and oxaloacetate from 2 h labeling experiments, we performed our next labeling experiments with ^13^C_1,2_-Glucose or ^13^C_5_,^15^N_2_-Glutamine for 12 h (with unsorted CA46 cells) to gain a better clarity of the carbon exchanges occurring in these reactions. In CA46 cells using ^13^C_1,2_-Glucose, we observed 50% of pyruvate (representing half of a molecule of glucose) was ^13^C-labeled, and nearly equal percentages of lactate and 45% of alanine (transaminase-derived) were also ^13^C-labeled (Figure 2b). Altogether we observed proportionally similar ^13^C labeling patterns in pyruvate, lactate, and alanine, at both 2 and 12 h labeling experiments. However, we were then able to detect 25% of α-ketoglutarate as ^13^C labeled (TCA cycle-derived) and 20% of glutamate with ^13^C label (transaminase-derived) (Figure 2b). Despite extending the duration for labeling, oxaloacetate did not show any evidence of ^13^C labeling, while aspartate showed <5% of glucose-derived ^13^C labels (Figure 2b). These results suggest that glucose carbons entering glycolysis are predominantly converted to lactate or undergo transamination into alanine (Figure 2b). The slower kinetics of ^13^C labeling in α-ketoglutarate suggests that only a small fraction of pyruvate enters the TCA cycle (Figure 2b).

With ^13^C_5_,^15^N_2_-Glutamine-labeled CA46 cells, we observed high amounts of glutamate consisting of either (40%) ^13^C_5_, ^15^N_1_, (45%) ^13^C_5_, N_0_, and (<5%) C_0_^15^N_1_, indicating that glutamine-derived carbon is being directly deaminated (40%) and returned from the TCA cycle (50%) (Figure 2b). Furthermore, since glutamate and α-ketoglutarate contain 90% of the carbons labeled with ^13^C_5_, glutamine appears to be the major source of carbons for TCA. Interestingly, by comparing nitrogen transfers to alanine based on the carbons derived from glucose (C_0,_ 60%) or glutamine (^13^C_5_, 40%), we observed that ^15^N was present only in 10% of carbon derived from glucose (C_0_) and completely absent in the alanine derived from ^13^C glutamine. This indicates that cytosolic glycolytic pyruvate was likely the source of carbon (through glutaminolysis, the TCA cycle, and transamination) (Figure 2b). The absence of ^15^N in the 40% fraction of ^13^C-derived alanine suggests that alanine transaminase activity involving glutamate-derived N_0_ could have occurred in a different subcellular compartment, possibly in the mitochondria. Of note, in comparing the 2 h with the 12 h labeling, we found a substantial reduction in ^15^N labeling in C_0_, which suggests alanine derived from glycolytic is consumed or lost. Although aspartate, pyruvate, and lactate showed trace amounts of glutamine-derived ^13^C (Figure 2b), the slower labeling kinetics at 12 h indicate that glutamine is not the major carbon source for these metabolites.

In summary, these results (from the 2 and 12 h labeling experiments) show that the carbon from glucose that ends in pyruvate is preferentially converted into lactate and alanine instead of directly being oxidized through the TCA. Thus, our next objective is to determine whether the pyruvate to alanine transamination that occurs during the S phase is related to differential glucose and glutamine utilization in normal and malignant cells. Therefore, we compared the metabolic profiles using LCL (lymphoblastoid, transformed non-malignant normal human B lymphocyte) and DLBCL cell lines (CA46 and SUDHL4).

We observed that in comparison with LCL, lymphoma cells (CA46 or SUDHL4) have significantly higher amounts of glucose-6-phosphate and glutamate (Figure 3a), which are the first metabolic intermediates of glucose and glutamine metabolism. Lymphoma cells had a one-fold higher pyruvate pool than LCL cells, while lactate and alanine pools were 2–4 times and 4-fold higher, respectively (Figure 3a). Although pyruvate is more readily available, our results indicate that malignant cells do not prefer to oxidize pyruvate by TCA, despite the higher availability. Therefore, the transamination of pyruvate to alanine represents a highly relevant aspect of Warburg’s paradox, which considers glucose as necessary for maintaining oxidative metabolism.

Based on these results, we concluded that by sparing glucose from oxidation, the transamination of pyruvate to alanine becomes crucial for sustaining the TCA cycle. This metabolic reprogramming likely provides a proliferative advantage to malignant cells. Therefore, our next goal is to investigate the overall metabolic implications of the differential utilization of glucose and glutamine between DLBCLs and LCL cells. By using ^13^C_1,2_-Glucose as tracer and performing 12 h labeling, we observed marked labeling of all intermediates in glycolysis, TCA cycle, transaminase, and nucleotide metabolism in all cell lines (Figure 3). We observed, however, that the proportion of labeling patterns varied between metabolic pathways and cell types (Figure 3). In non-malignant LCL cells, the average carbon labeling index (from ^13^C_1,2_-Glucose) for glycolytic intermediates beyond glucose-6-phosphate were 50% (representing half of a molecule of glucose), followed by 40% from citrate through isocitrate and 25% from α-ketoglutarate through the rest of the TCA cycle (Figure 3a). Additionally, fractional labeling with ^13^C derived from glucose was significantly higher for metabolic end-products representing nucleotides (60%) or coenzymes (NAD^+^ and NADP^+^) (80%) than for intermediates representing glycolysis or the citric acid cycle in LCL (Figure 3a). When comparing LCL with lymphoma cell lines CA46 and SUDHL4, the average percentage of ^13^C labeling with glycolytic intermediates was the same, but the labeling index for citric acid cycle intermediates, alanine, and nucleotides was 20% higher in lymphoma cell lines, CA46 and SUDHL4 (Figure 3a).

In parallel experiments using ^13^C_5_,^15^N_2_-Glutamine, we observed ^13^C labeling of all citric acid cycle intermediates, alanine from transaminase but not detected in the glycolytic intermediates, aspartate, and nucleotides in all three cell lines (Figure 3b). In comparison with LCL, the lymphoma cells, CA46 and SUDHL4, showed an average of 20% higher ^13^C labeling, with citric acid cycle intermediates and NAD^+^ and NADP^+^ (Figure 3b). Interestingly, when comparing LCL with lymphoma cells (CA46 and SUDHL4), we observed a net 20% increase in ^13^C incorporation from both glucose- and glutamine-derived carbons in α-ketoglutarate (Figure 3a,b). This suggests that glucose metabolism, while facilitating increased participation of glutamine, also increases. Consequently, glucose and oxidative metabolism are synergistically enhanced in lymphoma. Comparing these results with the similar 20% increases in glucose-derived nucleotide metabolism (Figure 3a), it becomes clear that the biological advantage of glucose-mediated (net gain of 40%) increases in oxidative metabolism could be significant for lymphoma. This helps meet the simultaneous demand for energy and glucose-derived nucleotides.

### 3.3. Molecular-Metabolic Circuitry of Lymphoma

Our findings indicate that pyruvate–alanine transamination may play a pivotal role in Warburg’s paradox. Thus, glucose, even when converted to lactate, synergistically favors oxidative metabolism to provide nucleotides and energy for the growth of malignant cells. Typically, during cell division, the process prioritizes acquiring nutrients and accumulating energy in the form of ATP [36]. This accumulation leads to feedback inhibition of glycolysis, which then permits nucleotide biosynthesis to proceed [36]. Due to the high energy demand for nucleotide metabolism, normal cells cannot use glucose for both energy and biosynthesis simultaneously [36]. However, malignant cells undergo metabolic reprogramming to overcome this limitation, resulting in increased glucose uptake but reduced glucose oxidation, without compromising oxidative metabolism, which is crucial for survival [1,2,5,24,37]. Therefore, Warburg’s paradoxical metabolism becomes crucial and must be considered a key endpoint in oncogenic paradigms that drive malignant proliferation. Disrupting these mechanisms in malignant cells could revert their metabolic and proliferative functions of normal cells. Therefore, our next objective is to construct a molecular metabolic circuit aligned with this metabolic paradox and then disrupt metabolic activities using inhibitors appropriate for selectively blocking cell proliferation in lymphoma, while sparing non-malignant LCL cells.

By enumerating the key metabolic genes that included pyruvate metabolizing enzymes and the associated metabolite transporters, defined as core metabolic components (Figure 4a), and then identifying the corresponding regulatory genes, we constituted a molecular-metabolic network model, as follows. Using GeneHancer, a database that contains genome-wide datasets containing promoter-gene associations, we conducted data mining and identified 2611 potential interactions consisting of 456 transcription regulators associated with 10 metabolic genes and transporters (see Appendix A). We then utilized transcriptomics datasets (see Appendix A), which included lymphoma patients (*n* = 481, from the National Cancer Institute) and lymphoma cell lines (*n* = 9) collected from our prior studies [31,33], to retrospectively analyze the mRNA level expression of metabolic genes, transporters, and transcription factors. The results of this analysis indicated that the medians of log_2_ normalized gene expression, for both lymphoma patient and cell line datasets, and were significantly high (log_2_ > 10 for metabolic genes and >7.5 for transcriptional regulators (Figure 4b)).

Among all genes, lactate dehydrogenase genes (LDHA and LDHB) were observed to be the most highly expressed (among the entire transcriptome), followed by transaminase genes (GOT1, GOT2, GPT, and GPT2) with log_2_ median expression > 10, both in lymphoma patients and cell lines (Figure 4b). In both lymphoma patients and cell lines, a median log_2_ of >10 and >8 was observed for transcriptional regulatory factors of metabolic genes (Figure 4b). Our next step was to isolate common sets of transcription factor-metabolic interactions via network analysis by Cytoscape, which identified 103 transcription factors (out of 456) that acted with eight metabolic genes as well as the PCNA associated with proliferation (Figure 4c). An analysis of pathway enrichment using the 103 transcription factors by the Gprofiler database revealed that these regulatory genes are associated with “Transcriptional misregulation in cancer”, KEGG:05202 pathway with the highest significance (adjusted *p* < 0.0001). Accordingly, this finding of a massive number of oncogenic transcription factors exerting redundant control over metabolic genes indicates the sheer complexity of the molecular mechanism employed in the reprogramming of metabolism in cancer. Moreover, such molecular complexity and redundant regulatory influences of metabolic genes present significant difficulties for the appropriate identification of actionable targets for biological investigation.

Nevertheless, we chose to assess these interactions by means of pharmacological inhibitors for a priori determination of the gene regulatory and enzymatic functions in the context of this molecular-metabolic circuitry. For this purpose, through the comparison of transcriptomes of lymphoma patients and cell lines, we identified 10 highly expressed transcription factors (median log_2_ > 10) that regulated a minimum of five metabolic genes (Figure 4c). Of note, despite shortlisting transcriptional factors to a few candidates, Figure 4c shows that each metabolic enzyme (at mRNA level) is regulated by multiple transcriptional factors and therefore requires additional screening experiments using a panel of pharmacological inhibitors.

### 3.4. Targeting the Molecular-Metabolic Regulatory Mechanisms in DLBCL

The overall goal of this experiment is to determine whether disrupting key molecular-metabolic circuits, thereby reversing the metabolic features associated with Warburg’s paradox, can limit the proliferative activity of malignant cells. Using the molecular-metabolic circuitry defined in Figure 4c, we selected the appropriate pharmacological inhibitors to compare cell proliferation assessments with LCL, CA46, and SUDHL4 cells. The inhibitors intended to target metabolic enzymes, such as SLC25A1 (CPT1) [38], MCT1 (MCT-III) [39], LDHA (oxamate) [40], glutamine transporter (MK801) [41], and PDH/α-ketoglutarate (Demvistat) [42], without affecting cell viability in the lymphoma cells or LCL (see Appendix A). While pharmacological inhibitors chosen for targeting transcriptional regulatory factors, besides TARDP1 (XAV939) [43], targeted HDAC1 (abexinostat) [44], STAT1 (fludarabine) [45], NONO (auranofin) [46], DNMT1 (thioguanine) [47], and ETS1 (YK-4-279) [48], all resulted in decreased cell viability in the lymphoma cell lines in a concentration-dependent manner, in both lymphoma and LCL cells (see Appendix A). Among these inhibitors, fludarabine alone selectively reduced the cell viability in CA46 and SUDHL4 lymphoma cell lines, without affecting LCL cell viability (see Appendix A). Therefore, for metabolomic profiling experiments, LCL, CA46, and SUDHL4 cells were treated with abexinostat (12.5 μM), fludarabine (2.5 μM), auranofin (0.5 μM), thioguanine (2.5 μM), and YK-4-279 (12.5 μM) for 48 h using the average IC_50_ for lymphoma cell lines.

Heatmap analysis of log_2_ transformed pool size estimates and hierarchical clustering of 41 significant metabolites (from ANOVA), revealed the following changes in metabolic signatures (Figure 5a), (Raw profiles and statistical analysis provided in Appendix A). Compared to lymphoma cell lines (CA46 and SUDLH4), LCL had a larger metabolic pool of glycolytic intermediates (glucose-6-phosphate, fructose-6-phosphate, dihydroxyacetone phosphate) and glutamate (Figure 5a). However, the pool sizes of nucleotides and TCA intermediates were higher in lymphoma cell lines than LCL. Treatment with fludarabine in CA46 and SUDHL4 resulted in reversing these metabolic features and became comparable to LCL. (Figure 5a). Specifically, fludarabine treatment, while increasing the levels of the glycolytic intermediates, caused significant decreases in metabolic pool sizes of pyruvate, lactate, TCA cycle intermediates, nucleotides, and alanine, selectively in the lymphoma cell lines (CA46 and SUDHL4 (Figure 5a). Abexinostat exhibited a significant reduction in the pool size of most metabolites in both LCL and DLBCL cells, while thioguanine and YK-4-279 demonstrated significant decreases in the pool sizes of all metabolites in LCL (Figure 5a). Auranofin did not show any significant difference in the metabolic profiles when compared with untreated cells in all cell lines (Figure 5a).

Additionally, we performed principal component analysis (PCA) for an unbiased assessment of the overall metabolic characteristics (of pool size changes) from each experiment. Based on the loadings from the PC1 axis (plotted by highest variation 56.7%), we observed that the overall metabolic pools in lymphoma cell lines CA46 and SUDHL4 are similar but differ from those in LCL (Figure 5b). Further, we observed that the metabolic profiles of fludarabine-treated CA46 and SUDHL4 were aligned together with fludarabine-treated LCL and untreated controls, suggesting that while LCL is unaffected by fludarabine, the metabolic profiles of fludarabine-treated lymphoma cells appear similar to those of LCL cells (Figure 5b).

A VIP scoring (see Methods), performed to identify the most significant metabolites which demonstrated most variations across all experiments, revealed lactate and TCA cycle intermediates, including α-ketoglutarate and nucleotides, as the most responsive metabolic signatures (Figure 5c). Since glucose and glutamine are the primary metabolites involved in all these metabolic processes, we also investigated the most significant changes in metabolic features by correlating the metabolic behavior with glucose from fludarabine treatment. The Spearman rank correlation of the significant metabolites showed positive correlations with glucose and glutamine and aspartate, whereas negative correlations were observed for lactate, alanine, TCA cycle intermediates, and nucleotide pool size changes (Figure 5c). The inverse correlation observed from treatment with fludarabine suggests that metabolic impairment, which decreases the TCA cycle, nucleotides, lactate, and transaminase metabolic intermediates, leads to the impaired utilization of glucose and glutamine, resulting in the corresponding increases. These metabolic changes affected proliferative function only in fludarabine-treated lymphoma cells, but not in the LCL cells (Figure 6a).

### 3.5. Fludarabine Interrupts Glucose Carbon Entry into Nucleotide Metabolism 

As summarized in Figure 6b, fludarabine treatment resulted in a significant decrease in the pool sizes of pyruvate, alanine, and lactate (Log 1-fold), as well as TCA cycle intermediates and nucleotide metabolites (Log 2-fold). On the other hand, the pool sizes of most glycolytic intermediates were increased (Figure 6b). Given that fludarabine treatment significantly reduced the levels of transaminase-associated pyruvate, alanine, and α-ketoglutarate, and meaningful insights from ^15^N labeling were not possible, we conclude that overall transamination is impaired, as evidenced by the significantly elevated glutamine levels. Based on these observations, our next objective is to identify which metabolic compartments of glucose and glutamine are affected and how they correlate with the decrease in pyruvate to alanine transamination caused by fludarabine. To achieve this, we performed the following isotopic tracer investigations using LCL and lymphoma cells.

Using ^13^C_1,2_-Glucose as a tracer, we examined the effect of fludarabine on the fractional labeling patterns with the metabolic intermediates in lymphoma and LCL cells, following 48 h of treatment with 12 h labeling investigations. With fludarabine treatment, although lymphoma cells showed increases in pool sizes of glycolytic intermediates, no significant changes were observed in the fractional labeling pattern, including lactate and alanine (see Appendix A). However, the most significant impact of reduced metabolic labeling with ^13^C_1,2_-Glucose was observed in the TCA cycle and nucleotide metabolic intermediates in the lymphoma cells treated with fludarabine, but not in the LCL cells (Appendix A). We further determined that, based on the carbon labeling patterns in the nucleotides, these nucleotides were primarily derived from glucose, through the oxidative pentose phosphate pathway (PPP) (Appendix A). The isotopic tracing of ^13^C_1_ incorporation in nucleotides, illustrated in Figure 6c and focusing on oxidative PPP, clearly demonstrates that fludarabine causes a significant decrease in ^13^C fractional labeling in all nucleotides (nucleotide mono- and triphosphates), only in the lymphoma cells (Figure 6c). Comparing the average ^13^C fractional labeling in the nucleotides, it is apparent that lymphoma cells incorporate far more ^13^C_1_ from glucose (65–70%) than LCL (40%) (Figure 6d). Meanwhile, treatment with fludarabine resulted in a significant reduction (from 65–70% to 10–15%) in glucose-derived ^13^C_1_ labeling of nucleotides in the lymphoma cells, compared to LCL (Figure 6d).

Based on the estimation of relative carbon contribution between oxidative and non-oxidative PPP, we observed that lymphoma cells utilized oxidative PPP (70–80%) and LCL (50%), with fludarabine treatment resulting in the most reduction in oxidative PPP (from 70–80%) to 30% in lymphoma compared (from 50%) to 40% in LCL versus non-oxidative PPP (Appendix A). These results indicate that the oxidative pentose phosphate pathway (PPP) is a major source of nucleotides and is elevated in lymphoma. Additionally, fludarabine treatment restores the oxidative PPP to levels comparable with those in LCL. Based on the relative carbon contribution estimation, lymphoma cells utilize more glucose-derived carbons (20–30%) for ribose phosphate synthesis (R5P) than LCL (18%), which decreases to less than 10% in fludarabine-treated lymphoma cells (Figure 6e). However, the contribution of glucose carbon to alanine and citrate was similar in LCL cells (Figure 6e), while the contribution to lactate could be different, since both intracellular and secreted lactate assessments are necessary for making meaningful conclusions. Despite pyruvate, lactate, and alanine fractional labeling patterns remaining unchanged, their pool sizes showed significant reductions (Figure 6f). However, there was also a significant increase in upstream glycolytic metabolites glycerol-3-phosphate and dihydroxy acetone phosphate (DHAP) in fludarabine-treated lymphoma cells (Figure 6f), indicating that overall glycolytic activity, if not inhibited, is accumulating.

Taken together, fludarabine treatment reduced the metabolic pool sizes of nucleotides, pyruvate, lactate, and alanine in lymphoma cells, with opposite effects on upstream glycolytic intermediates, as summarized in Figure 6g. This was accompanied by a decrease in the incorporation of glucose-derived carbon into the TCA cycle and nucleotide metabolism. The correlation between lactate and alanine levels with energy metabolism and nucleotide levels is therefore consistent with the fludarabine-mediated suppression of lymphoma specific cellular proliferation.

### 3.6. Fludarabine Disrupts Glutamine Carbon Entry into the TCA Cycle

Next, we investigated whether reduced pyruvate availability is linked to elevated glutamine levels and the decreased metabolic activity of the TCA cycle (Figure 6b). This was conducted to establish that glucose, even when converted into lactate, acts synergistically with oxidative energy-yielding metabolism. As illustrated in Figure 7a, we investigated ^13^C enrichment patterns of TCA cycle intermediates using glutamine-derived ^13^C-5 glutamate as a precursor for this investigation. Oxidation of α-ketoglutarate in the conventional direction of the TCA cycle should result in the loss of one carbon as carbon dioxide generating ^13^C-4 succinate, which in subsequent metabolic stops could yield ^13^C-4 citrate, through ^13^C-4 oxaloacetate accepting two carbons from pyruvate. Alternatively, if α-ketoglutarate becomes the direct source of citrate through reductive carboxylation [12], then all five carbons of citrate will be derived from ^13^C-5 sourced from α-ketoglutarate, as shown in Figure 7a.

Using ^13^C_5_,^15^N_2_-Glutamine as a tracer, we observed that C-5 of α-ketoglutarate, aconitate and citrate were labeled at 80%, 50%, and 40% respectively in the untreated lymphoma cells (Figure 7b). In LCL cells, that C-5 of α-ketoglutarate, aconitate and citrate was labeled with ^13^C at 50%, 38%, and 18% respectively from ^13^C_5_,^15^N_2_-glutamine. Together, these labeling patterns indicate that glutamine significantly contributes to citrate synthesis through reductive carboxylation in lymphoma than LCL cells. Considering the loss of one carbon atom as CO_2_ in the conventional TCA cycle, we found that succinate C-4 is 70% labeled, while C-4 of fumarate and malate are labeled at 40% (as shown in Figure 7b). Unfortunately, due to its low abundance, the labeling in oxaloacetate remained undetermined by mass spectrometry. Taken together, our ^13^C enrichment results reveal that glutamine carbon undergoes both oxidation (via the conventional TCA cycle) and reductive carboxylation generating citrate (Figure 7b). This necessity for (glutamine-derived) excess citrate production suggests that citrate is lost from the TCA cycle or is not sufficiently synthesized from pyruvate-derived Acetyl-CoA. Our previous research has shown that citrate exit facilitates nucleotide metabolism (via glucose and the PPP) [24]. Therefore, it is possible that glutamine-derived citrate could be necessary for compensating for this citrate loss. Taken together, we conclude that citrate exit is another important link between energy metabolism and nucleotide synthesis.

While fludarabine treatment did not alter the proportion of ^13^C enrichment in C-5 of α-ketoglutarate, citrate, and C-4 of succinate in LCL cells, these ^13^C enrichments, except for citrate, were significantly reduced in the lymphoma cells (CA46 and SUDHL4) (Figure 7c). However, we observed a significant decline in the citrate pool size in the fludarabine-treated lymphoma cells (CA46 and SUDHL4), along with α-ketoglutarate and succinate (Figure 7d). This abrupt decline in citrate occurred despite increases in the pool sizes of glucose-6-phosphate (Figure 6f) and glutamine (Figure 7d), indicating that citrate is the end-product of the synergistic glucose/glutamine metabolism in the TCA cycle.

Furthermore, by comparing the relative contribution of carbons ^13^C-1 from glucose and ^13^C-5 or ^13^C-4 from glutamine, we observed that lymphoma cells incorporate 20% more carbon from glutamine into citrate (Figure 7e). The relative contributions of carbons from glucose and glutamine to α-ketoglutarate, succinate, and malate were similar for both LCL and lymphoma cells (Figure 7e). However, treatment with fludarabine further reduced glucose carbon contributions to α-ketoglutarate and succinate (from 30–40% to less than 10%) (Figure 7e), indicating that glutamine entry facilitates the reciprocal participation of glucose carbon in the TCA cycle.

Taken together, results from the fludarabine treatment experiments indicate that decreased pyruvate and alanine availability (Figure 6f) restrict the participation of both glucose and glutamine carbons in the TCA cycle, resulting in glycolytic and glutamine accumulation and thereby resulting in the reduction in the pool sizes of TCA cycle intermediates (Figure 7d). Therefore, pyruvate to alanine transamination appears to be essential for the glucose-dependent maintaining of TCA cycle metabolism through glutamine.

### 3.7. Elevated Levels of Pyruvate, TCA Cycle Intermediates, and Nucleotides Correlate with STAT1 Expression in Tumors

Our experiments with fludarabine have demonstrated that when glucose and glutamine carbons accumulate and fail to enter the energy-yielding TCA cycle, nucleotide biosynthesis becomes compromised. These observations are significant in the context of Warburg’s paradox, where glucose, despite being converted into lactate, acts synergistically with oxygen metabolism to promote tumorigenesis. Therefore, Warburg’s paradox and the Warburg effect represent the endpoints for oncogenic signals to generate energy and nucleotides, driving malignant cell proliferation. Our final goal is to determine whether the metabolic correlations inferred from in vitro experiments using cultured cells are clinically relevant for lymphoma. Our comparison of the metabolic profiles of cultured cells (LCL and lymphoma cell lines, CA46 and SUDHL4) with patient-derived tissues (normal lymph nodes and B cell lymphoma tumors, *n* = 10) reveals that the pool sizes of metabolic intermediates from transaminase, the citric acid cycle, and nucleotide metabolism are well correlated and consistently elevated in malignancy (*p* < 0.05), with significant metabolites shown as a heatmap in Figure 8a,b. Raw profiles and detailed statistical analysis are provided in Appendix A.

Additionally, we compared protein lysates from tumor and normal tissues for expression levels of metabolic enzymes and transcription factors associated with our predicted Warburg circuitry (Figure 4c) for validating the biological appropriateness for using fludarabine in our metabolic studies. The results of the Western blot analysis comparing normal lymph nodes to B cell lymphoma nodes indicated that STAT1 and JUND are significantly overexpressed in lymphomas (Figure 8c). We also observed that the expression of LDHA and alanine transaminase in lymphoma is sporadically elevated. The significant overexpression of STAT1 (Figure 8c) indicates that fludarabine’s tumor selectivity is associated with its regulatory influence on inhibiting STAT1 function and malignant metabolic activity, and it is clinically relevant.

In summarizing the results of our experiments, we observed an increase in pyruvate, lactate, and alanine levels during the S phase of the cell cycle, which corresponded to increases in glycolysis, the TCA cycle, and nucleotide metabolism (Figure 1). Pyruvate plays a key role in supplying glutamine-derived carbons as α-ketoglutarate and contributes to citrate formation in malignant cells (Figure 2 and Figure 7). We previously observed that citrate exit facilitates FASN-mediated NADP/NADPH cycling, which is necessary for diverting glucose-derived carbon into nucleotide synthesis via the oxidative PPP [24]. Therefore, continuous citrate loss should result in a diminished TCA cycle, as shown in step 1 (Figure 9a). This is evident from the shrinking citrate pools derived from glucose and glutamine with fludarabine treatment (Figure 9a).

Transamination of glycolytic carbons from pyruvate to alanine serves as an alternate pathway to restore the TCA cycle (and citrate) through glutamine carbon, as indicated in step 2 (Figure 9a). This is evident from the decreases in α-ketoglutarate levels and label incorporations observed with fludarabine (Figure 7c,d). As the entry of pyruvate carbon into the TCA cycle is diminished with fludarabine, the malate, aspartate, and oxaloacetate shuttle shunts glutamate carbons through a partially active TCA cycle, as indicated in step 3 (Figure 9a), potentially sustaining oxidative metabolism. With the accumulation of the metabolic components of mitochondrial shuttle pathways—such as aspartate (Figure 7d), DHAP, and glycerol-3-phosphate (Figure 6f)—with fludarabine, we conclude that the overall integrity of the TCA cycle is vulnerable. Therefore, constant availability of pyruvate becomes crucial for TCA cycle to remain operational, and metabolic conversion of pyruvate to lactate ensures this availability, as shown in step 4 (Figure 9a).

Finally, based on the fact that fludarabine treatment decreases pyruvate and lactate levels while glycolytic intermediates accumulate, the impaired TCA cycle-mediated energy metabolism is accompanied by decreased nucleotide biosynthesis, as indicated in step 5 (Figure 9a). This conclusion is derived from Figure 6g and Figure 7f. We conclude that all these metabolic reprogramming events converge on nucleotide biosynthesis to confer a proliferative advantage in malignant lymphoma cells.

Applying these results in the context of resolving the metabolic reprogramming principles responsible for the Warburg effect and Warburg’s paradox, and their implications in malignancy, our conclusions are as follows:

The physiological regulation of normal cell proliferation is a systematically controlled process facilitated through feedback regulations that follow a sequential metabolic activity [36]. Newly formed cells prioritize glucose carbon for energy production. The accumulating energy then exerts feedback inhibition over pyruvate synthesis [49], resulting in the accumulation of glycolytic intermediates. This allows glucose carbons to flow toward nucleotide biosynthesis. Once nucleotide levels reach sufficient levels, mitotic activity is triggered followed by completion of cell proliferation in an orderly fashion [36] (Figure 9b).

The hallmark of malignancy, on the other hand, is uncontrolled cell proliferation. The metabolic feedback regulations found in normal cells make it disadvantageous for cells to commit to rapid proliferation. Therefore, the overexpression of LDH pivots glycolysis away from TCA cycle-mediated feedback inhibition, keeping the glucose carbon flow uninterrupted. This is the foremost and most apparent feature noticeable in most malignancies and caught the attention of Warburg, commonly known as the Warburg effect [2] (Figure 9b).

With LDH activity keeping pyruvate metabolism open, transamination reactions allow coupling of glucose with glutamine to sustain the TCA cycle and respiratory oxygen-mediated energy metabolism (Figure 9b). Together these metabolic features constitute the principles behind Warburg’s paradox. As a result, with excess energy constantly available, increased glycolytic and glutamine metabolism via the TCA cycle allows citrate to couple with FASN-mediated Ox-PPP-dependent nucleotide synthesis [24]. FASN is also the most highly overexpressed protein in almost all malignancies, including in lymphoma [24] (Figure 9b). Considering that exogenous lipids are physiologically abundant, de novo lipogenesis in malignancy, which is energetically expensive, is unnecessary but exists [24]. Additionally, citrate, being the negative feedback regulator of the TCA cycle, is utilized for lipogenesis [49], providing an opportunity to remove another important feedback regulatory mechanism over energy production. LDH and FASN are crucial metabolic functions that remove feedback restrictions on glycolysis and the TCA cycle, enabling citrate to integrate energy and nucleotide metabolism. Due to the low abundance of oxaloacetate and the contribution of glutamine-derived carbon in the reductive synthesis of citrate, it is clear that the exit of citrate is also a crucial component for integrating glucose and oxidative metabolism. Consequently, all oncogenic signals ultimately converge on upregulating the expression of enzymes involved in these metabolic paradigms to drive abnormal cell proliferation.

## 4. Discussion

Based on Warburg’s concept, the synthesis of lactate from glucose suggests that glucose oxidation is unnecessary (Warburg effect) [1]. However, the simultaneous need for both glucose and oxygen in malignancy in the context of lactate metabolism though seemingly contradictory, indicates that these metabolic functions as complementary mechanisms (Warburg’s paradox) [1]. To understand the broader biological implications of the Warburg effect in cancer, it is essential to explore the metabolic impact of these paradoxical features.

Although extensive research on the Warburg effect exists, there is a lack of a simplistic overview and clarity, leading to several unanswered questions raised in the literature [3,4,9]. Despite ongoing debates about the biological benefits of aerobic glycolysis in malignancy [49], we independently assessed this using multiple experimental approaches to understand its overall relationship with lymphoma metabolism. This study aims to examine and clarify the metabolic patterns in lymphoma and non-malignant cells, investigate their behavior in malignant-specific proliferative activity, and illustrate metabolic connectivity using an ‘omics’-based strategy. While comparing absolute quantities between metabolites, flux analysis, subcellular compartmentalization, kinetics, and accounting for metabolite excretion are important next steps, our ‘omics’-based approach focuses on metabolic labeling patterns and relative changes in each metabolite under different conditions. This approach provides an initial overview of the metabolic reprogramming principles and its relevance to Warburg phenomenon in lymphoma.

Results from our metabolomic profiling and analysis of cell cycle progression in CA46, show that increased amino acid pools correspond to the G1 and S phases, and that glycolysis, TCA cycle, and nucleotide pools also increase during the S phase progression, with glycolysis and nucleotides remaining elevated until G2. While similar metabolomic profiling of cell cycle phases has been conducted with yeast, mouse fibroblasts and lymphocytes, such an assessment with cancer cell lines has proven to be very challenging [50]. Therefore, results from our metabolic profiling and isotope enrichment analysis demonstrating increased transamination of pyruvate to alanine occurring during S phase of the cell cycle, are significant findings in this study. Most importantly, ^13^C enrichments demonstrated that pyruvate and lactate carbons are derived entirely from glucose, while alanine had carbons derived both from glucose and glutamine in lymphoma. Interestingly, studies based on glioblastoma indicated that glutamine is the major source of carbon for pyruvate, alanine, lactate, and nucleotides [51,52], which suggests that these metabolic features while conserved, may vary by the source of carbon based on the cell type.

Based on the 2 vs. 12 h incorporation of ^13^C and ^15^N labels into alanine (derived from glutamine), we observed that the ^15^N label was present only in the C-0 isotopomer, which originates from glucose and occurs rapidly, as shown in Figure 2a. Conversely, alanine containing glutamate-derived ^13^C did not include ^15^N; it was identified as N-0, which followed slower kinetics (Figure 2a). These findings suggest that transamination reactions involving glucose-derived pyruvate and ^13^C,^15^N-glutamate occur in distinct cellular compartments. It is well known that cytosolic alanine transaminase converts glycolytic pyruvate to alanine, while mitochondrial alanine transaminase converts alanine to pyruvate for gluconeogenesis [53]. Moreover, transamination reactions are freely reversible metabolic functions [53]. Therefore, we conclude that the source of alanine containing ^15^N is glycolytic pyruvate. The resulting carbon exchange, yielding α-ketoglutarate from glutamate, occurs in the cytosol, directly linking glucose to the continuity of oxidative metabolism, as envisioned in Warburg’s paradox.

Further evidence in the literature supports our conclusion that, despite glucose being metabolized into lactate, glycolysis in malignant cells seems to catalytically enhance oxidative metabolism through the transamination of pyruvate to alanine, as elaborated further. Inhibition of alanine transaminase with cycloserine or 3-chloroalanine while reducing alanine levels, has also been shown to disrupt glycolysis and energy metabolism in Lewis lung carcinoma cells [54]. Similarly, blocking alanine or aspartate transaminase activities with oxamate has been shown to kill cancer cells while not affecting normal cells [54,55,56]. This effect is comparable to the selective inhibition of lymphoma proliferation observed in our studies with fludarabine treatment which corresponded with a reduction in pyruvate and alanine levels, compromising the TCA cycle and nucleotide metabolism. These results altogether illustrate the importance of transamination in the metabolic interplay between glycolytic pyruvate and oxidative metabolism as a significant event in the proliferative activities of lymphoma. Similar results have already shown that pyruvate to alanine transamination is a key metabolic link between glycolysis and the TCA cycle [57]. Interrupting this link using RNAi against cytosolic alanine transaminase (GPT2) was demonstrated to impair oxygen metabolism, cell proliferation, and tumor growth in colon cancer [57]. However, this study did not further elaborate on how these metabolic features are interconnected with proliferative functions, specifically nucleotide biosynthesis, which is critical for malignancy [57]. Moreover, inhibiting aspartate transaminase activity impedes the flow of glucose carbons into nucleotide metabolism and diminishes both the TCA cycle and oxygen consumption in MDA-MB-231 cells [56]. With proliferation blocked from aspartate transaminase in MDA-MB-231, this study reported that specifically, the carbon flow from glucose (as α-ketoglutarate) to glutamate was impaired [56]. Therefore, it is possible that both alanine and aspartate transaminases are tandemly participating in this breast cancer cells [56], while this dependence could be a predominant feature of alanine transaminase metabolism in lymphoma (this study) or colon cancers [57].

Given that nucleic acids constitute approximately one-third of a cell’s dry weight, glucose is essential, not only as an energy source but also for providing ribose for nucleotide biosynthesis, which is crucial for cell division [11]. Results from our experiments and others with stimulated peripheral lymphocytes [58], have shown that glucose utilization and lactate production surge during nucleic acid synthesis and S phase progression, suggesting that metabolic reprogramming of diminishing glucose oxidation to facilitate nucleotide biosynthesis may be advantageous for the proliferative functions of malignant cells. In this context, while other studies have addressed parts of this issue, our results provide an integrated overview showing that the Warburg effect and paradoxical metabolism are directly interlinked for simultaneous energy production and nucleotide biosynthesis. Meeting these two most fundamental necessities is important for malignancy, as this distinguishes it from normal cell proliferation, where these metabolic activities proceed in an orderly fashion.

We have observed that lymphoma cells cultured in a medium with dialyzed serum lacking glucose undergo rapid cell death within 2 h. Moreover, lymphoma cells (SUDHL4) can sustain proliferation with 1 µg/mL glucose per million cells (compared to the standard culture concentration of 100–200 µg/mL) Therefore, considering that physiological deprivation of glucose is not a practical therapeutic approach, targeting metabolic vulnerabilities requires identifying molecular or metabolic inhibitors and developing appropriate combinatorial therapies. In terms of translating these results into metabolism-targeted therapies, evidence suggests that fludarabine, commonly used for hematological malignancies, inhibits STAT1 activity [59,60,61]. Previous studies have established that STAT1 knockdown results in downregulation of genes associated with glycolysis, TCA cycle, oxidative phosphorylation and LDH [62]. However, in chronic lymphocytic leukemia patients, the effects of fludarabine on STAT1 were transient, leading to increased STAT1 expression and therapeutic resistance [59]. Existence of such complexity further underscores the need for developing comprehensive strategies that may involve drug combinations to simultaneously target multiple molecular, metabolic, and proliferative pathways when applying such concepts in cancer treatments.

In summarizing our investigations into the Warburg paradox, we have observed that pyruvate, lactate, and alanine surge during the S phase of the cell cycle (Figure 1b). This correlates with the entry of α-ketoglutarate from glutamine and the synthesis of nucleotides from glucose-derived carbon (Figure 2 and Figure 3). These metabolic features are elevated in malignant lymphoma cells compared to non-malignant LCL cells (Figure 3). Fludarabine, which inhibits cell proliferation in lymphoma cells but not in non-malignant LCL cells, perturbs these metabolic functions (Figure 5 and Figure 6). Our observations with fludarabine treatment also reveal that metabolic intermediates representing the TCA cycle and nucleotides are depleted only in lymphoma cells (Figure 6), likely due to metabolic consumption linked to oncogenic proliferative activities. Elevated glycolytic intermediates under these conditions correlate with a lack of nucleotide synthesis (Figure 6a), supporting our previously established role of TCA cycle-derived citrate in PPP activity.

Depletion of other TCA cycle intermediates aligns with their established roles in driving oncogenic activities, such as chromatin modification and proliferative signaling [22]. We have previously reported that TCA cycle gene elevation occurs early in tumor progression [31]. Thus, funneling TCA cycle intermediates for oncogenic proliferation appears to be a metabolic necessity of malignancy [22]. Due to these metabolic changes, with reduced oxaloacetate (the substrate for glucose oxidation), pyruvate takes alternate routes. Overexpression of LDH in malignancy leads to pyruvate being metabolized into lactate, a key phenotype in the Warburg effect [3]. Continuous lactate drainage prevents glycolysis from inhibiting glucose uptake, making glucose uptake a prominent feature that is apparent in cancer diagnosis [7,8].

Our findings also reveal that the metabolic conversion of pyruvate to alanine, catalyzed by a reversible transaminase reaction, allows glutamine carbon to feed as α-ketoglutarate while yielding alanine, and it is noteworthy that alanine is also a precursor for protein and nucleotide biosynthesis. Fludarabine-induced increases in metabolites associated with glycerol-3-phosphate and DHAP shuttle mechanisms (Figure 6f) which are elevated in cancer, known to sustain oxidative phosphorylation, [63] possibly continues to maintain the continuity of oxygen metabolism through glycolytic intermediates, while TCA intermediates remain depleted (Figure 6a). Based on glutamine carbon labeling in the TCA cycle, the bidirectional split toward α-ketoglutarate carbons into citrate and malate (Figure 7e) suggests that the TCA cycle is no longer active as conventional cyclical metabolism. This TCA metabolic fidelity loss, if occurring in mitochondria, may also represent a mitochondrial defect as envisioned by Warburg. While the existence of robust shuttle mechanisms that maintain oxygen metabolism could compensate for TCA defects [63], supplying glutamine-derived α-ketoglutarate, which make it synergistic with glycolysis, could explain the mechanistic basis of Warburg’s paradox.

In comprehending the translational significance by understanding the Warburg paradox, it appears that targeting oncogenic metabolic functions that participate in simultaneously enhancing carbon-feeding biosynthetic activities and oxygen-dependent energy-yielding processes is necessary for achieving superior efficacies.

## 5. Conclusions

The relationship between the Warburg effect, metabolic reprogramming, and cancer is a well-studied aspect of cancer biology. However, many questions and challenges remain about the direct biological and translational implications of this metabolic phenomenon in cancer [4,64]. Our results demonstrate that pyruvate transamination, coupling glycolysis, and glutamine sustain oxidative energy metabolism, aligning with the Warburg paradox. Producing lactate is essential to protect glycolysis from TCA cycle-mediated feedback inhibition. In return, the TCA cycle provides citrate, facilitating lipogenesis and coupling with the oxidative pentose phosphate pathway (ox-PPP) to produce nucleotides. This integrated perspective underscores the dual role and biological significance of the Warburg paradox in ensuring a continuous, uninterrupted supply of energy and nucleotides for malignant cell proliferation.

## Figures and Tables

**Figure 1 cancers-16-03606-f001:**
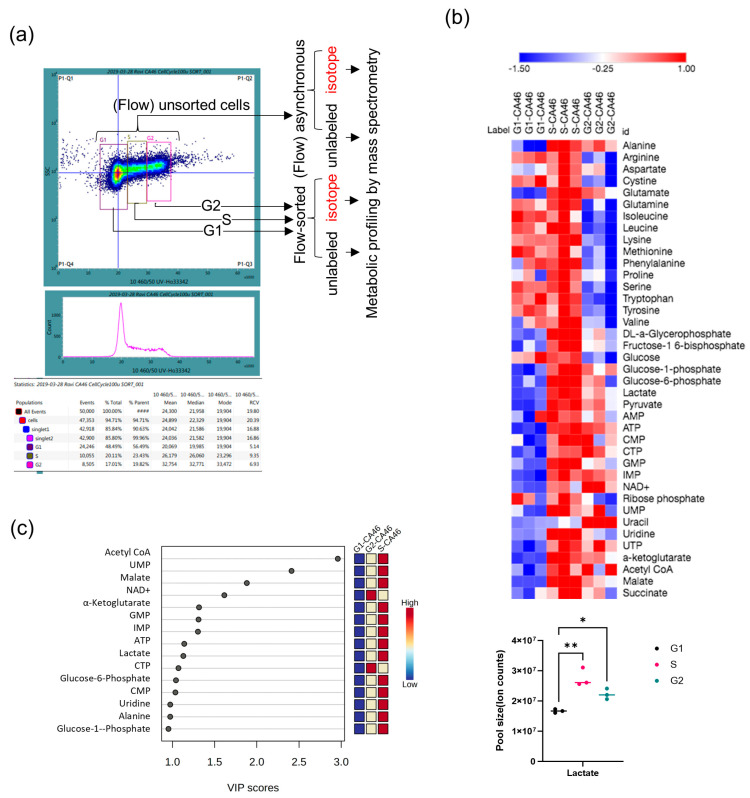
**Metabolomic profiling of lymphoma cell cycle.** (**a**) A representative flow cytometric profile of Hoechst 33342-stained CA46 cells illustrating the gating strategy adapted for sorting cells according to their cell cycle phase for metabolomic profiling. (**b**) Heatmap of metabolic pools profiled from mass spectrometry analyses of CA46 cells sorted by phase of the cell cycle. The color gradient represents the absolute mean deviations between low and high levels of each metabolite’s pool sizes. Statistically significant differences from the comparison between G1 with S or G2 are denoted (*, ** with *p*-values of <0.05, <0.005, respectively (**c**) Identification of top significant metabolite features (*p* < 0.001) by partial least squares—discriminant analysis and variable importance in projection scoring analysis. The colored boxes on the right indicate the relative concentrations of the corresponding metabolite in each group.

**Figure 2 cancers-16-03606-f002:**
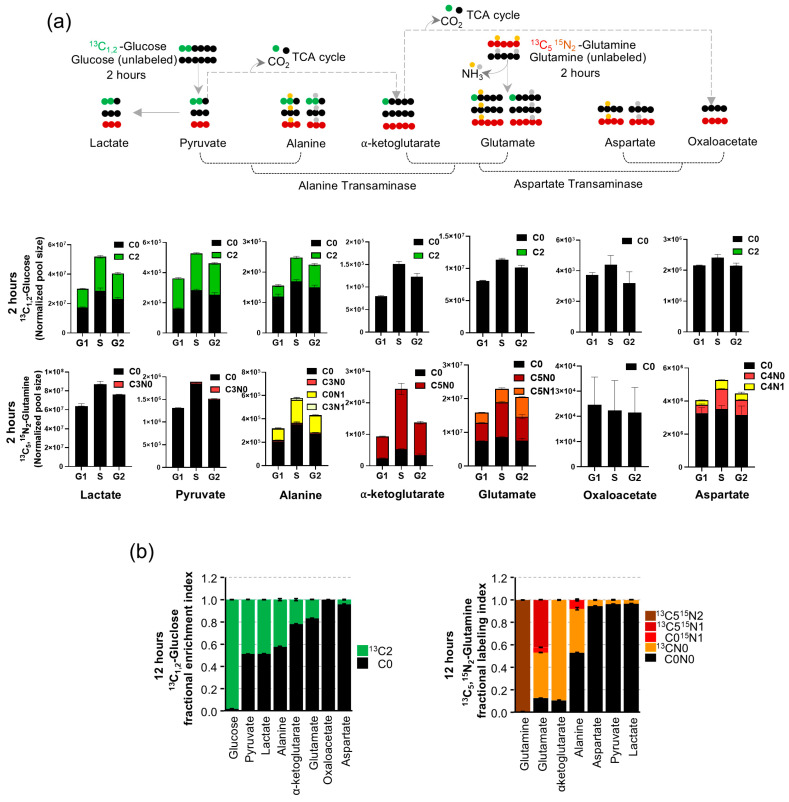
**Pyruvate and alanine transamination activity in cell cycle progression.** (**a**) Schematic diagram illustrates isotopomers arising from carbon and nitrogen exchanges in transaminase reactions and pyruvate metabolism. Bar graphs represent the pool sizes and isotope enrichment patterns from ^13^C_1,2_-Glucose or ^13^C_5,_^15^N_2_-Glutamine tracers from 2 h labeling with flow sorted CA46 cells used in the identification of metabolic changes associated with cell cycle progression. (**b**) Bar graphs represent fractional labeling patterns detected in the transaminase metabolism from 12 h labeling with ^13^C_1,2_-Glucose or ^13^C_5,_^15^N_2_-Glutamine tracers in CA46 cells.

**Figure 3 cancers-16-03606-f003:**
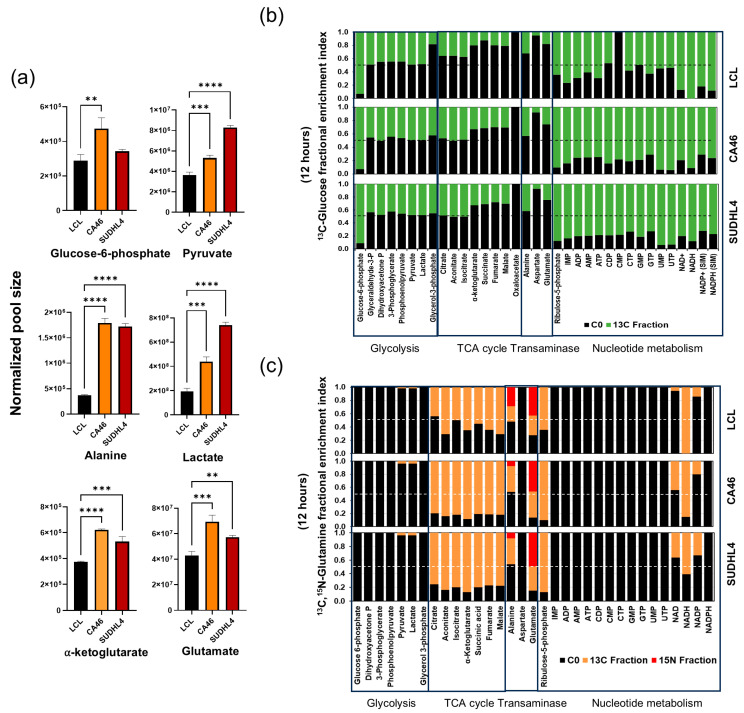
**Comparative analysis of isotope enrichment patterns in glucose and nucleotide metabolism in lymphoblastoid cells and lymphoma cell lines.** (**a**) Bar graphs represent relative pool sizes of transaminase metabolites in LCL vs. lymphoma cells. The error bars represent the standard deviation from the mean of experimental triplicates. Statistically significant differences from the comparison between LCL with lymphoma cell lines (CA46 or SUDHL4) are denoted (**, ***, **** with *p*-values of <0.005, <0.0005, and <0.0001, respectively), by 2-way ANOVA, as differences in the metabolite pool sizes between normal versus lymphoma cells. (**b**,**c**) Bar graphs represent mean fractional isotope enrichment patterns in glycolysis, citric acid cycle, transaminase, pentose phosphate, and nucleotide metabolism in LCL, CA46, and SUDHL4 with (**b**) ^13^C_1,2_-Glucose or (**c**) ^13^C_5,_^15^N_2_-Glutamine isotope tracers. Each bar represents the mean of experimental triplicates.

**Figure 4 cancers-16-03606-f004:**
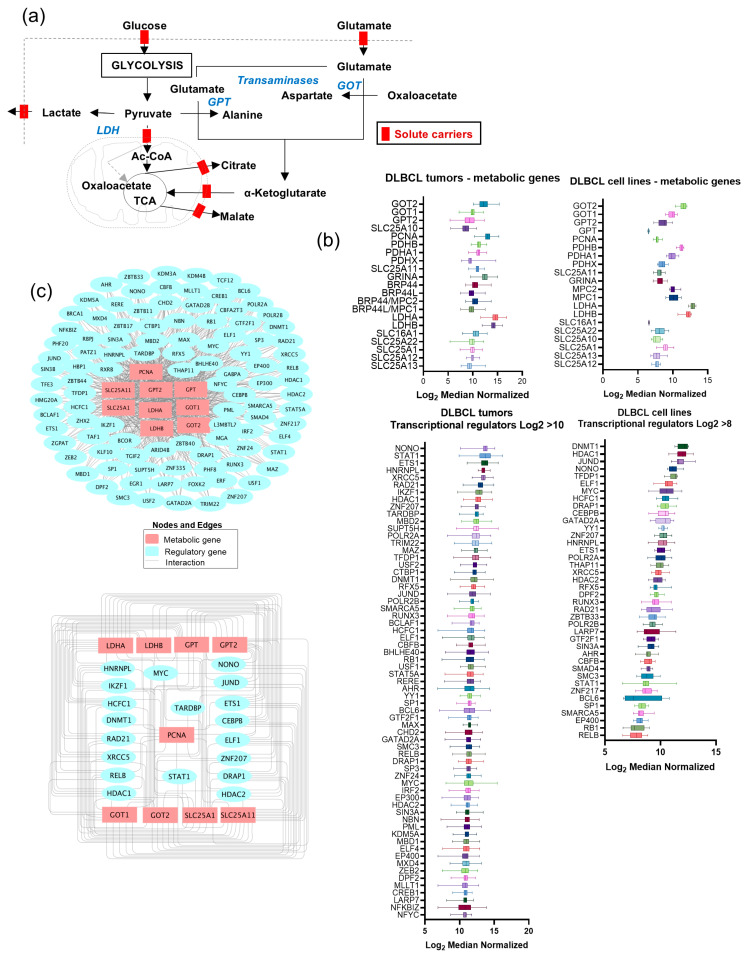
**Modeling the molecular metabolic circuitry of lymphoma.** (**a**) This diagram illustrates the metabolic intersections linking pyruvate with lactate, transaminase, the TCA cycle, and the associated solute carrier transporters. As described in the Methods Section, these metabolic components were used to model the molecular metabolic circuitry of the Warburg effect. (**b**) The boxplots illustrate the mRNA expression of metabolic genes and their putative regulatory factors in lymphoma tumors (*n* = 481) and cell lines (*n* = 10). Whiskers represent standard deviations from the normalized mean of Log_2_ expression values for each experimental dataset. (**c**) Molecular-metabolic network rendering using Cytoscape shows metabolic genes with putative regulators as interactors (**top**), further filtered by putative regulators with four or more interactions with metabolic genes (**bottom**).

**Figure 5 cancers-16-03606-f005:**
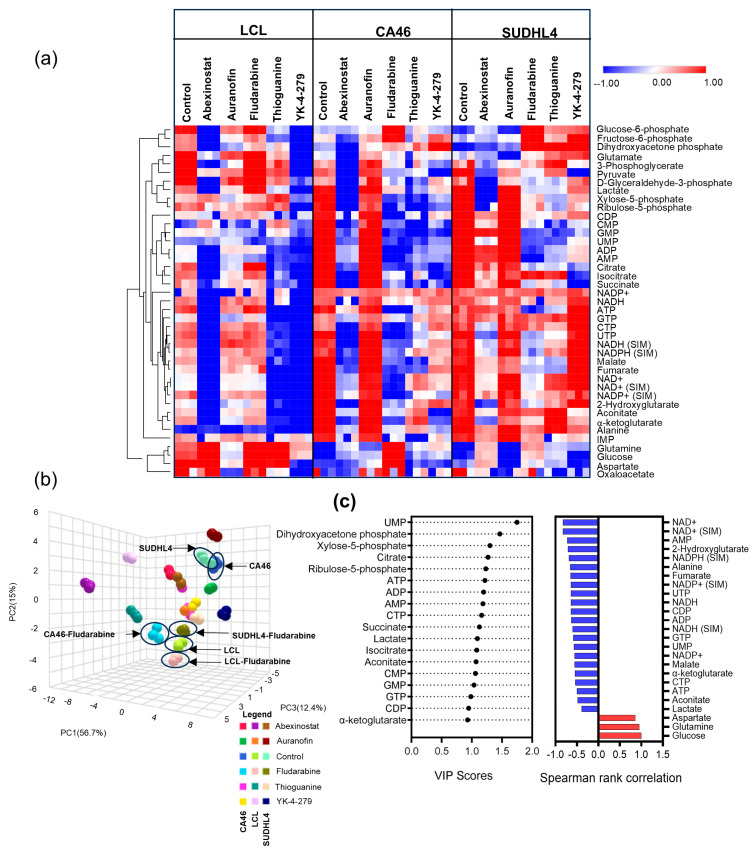
**Perturbation of molecular metabolic network with pharmacological inhibitors.** (**a**) The heatmap from metabolomic profiling shows the effect of pharmacological inhibitors on targeting Warburg metabolism regulators in LCL, CA46, and SUDHL4 cells. Color gradient indicates the absolute mean deviation between pool sizes for each metabolite. (**b**) Plots from the principal component analysis of LCL, CA46, and SUDHL4 demonstrate overall behaviors of each experimental set from treatment with pharmacological inhibitors intended to target the Warburg metabolism. (**c**) Partial least squares-based analysis (**left**) identifies the top significant metabolite features by discriminant analysis and variable importance (**right**) and metabolites aligned with changes in glucose metabolism with fludarabine based pharmacological inhibition of Warburg regulators by Spearman rank correlation analysis with *p*-values < 0.005 (**right**).

**Figure 6 cancers-16-03606-f006:**
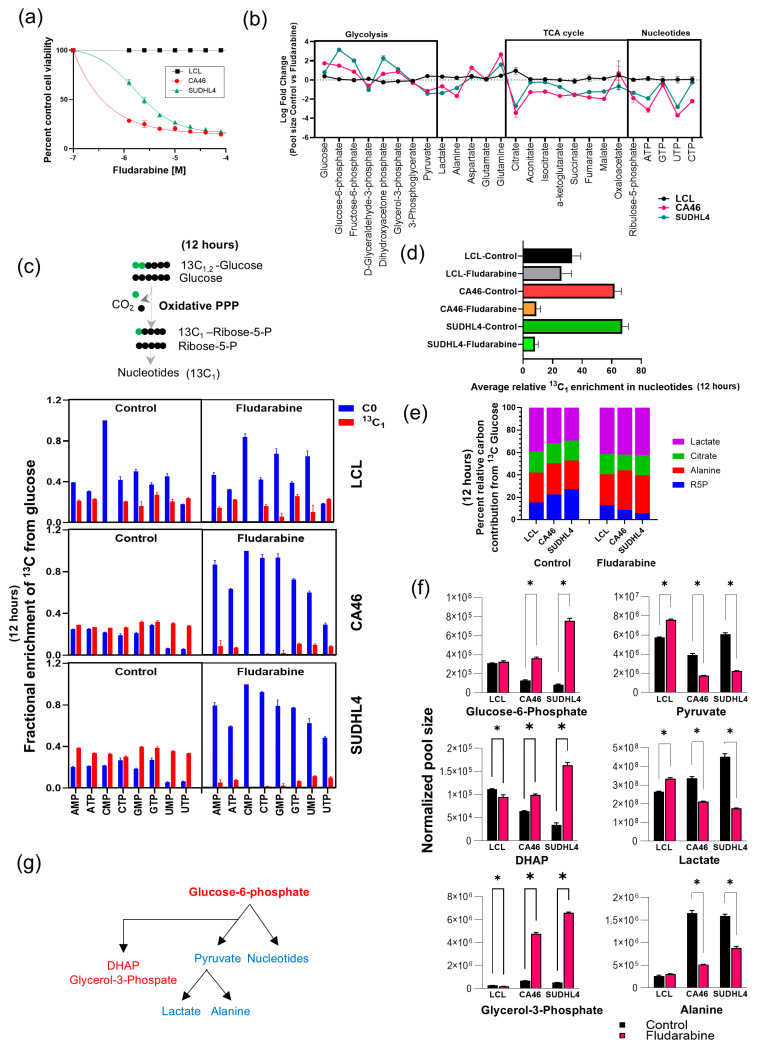
**Fludarabine decreases ^13^C incorporation from glucose into nucleotides.** (**a**) Dose–response curves from fludarabine lymphoma and LCL cells, with log molar concentration (x-axis) and percent cell viability (y-axis), determined at 72 h by CellTitre glo assay. (**b**) The line graph represents the mean of log_2_ fold-changes in the pool sizes of corresponding metabolites in LCL, and lymphoma cells, treated with fludarabine. (**c**) Schematics represent isotope enrichment pattern for ribose phosphate and synthesis of nucleotides derived from glucose through oxidative pentose phosphate pathway (Ox-PPP). Bar graphs represent mean fractional ^13^C enrichment patterns in nucleotides of control and fludarabine-treated LCL, CA46, and SUDHL4 cells. (**d**) Bar graphs represent percentages of average relative carbon contributions from ^13^C enriched nucleotides of control and fludarabine-treated LCL, CA46 and SUDHL4 cells. The error bars represent the standard deviation from the mean of all nucleotides. (**e**) Distribution plot represents relative carbon contributions from ^13^C enrichments in lactate, citrate, alanine, and ribulose-5-phosphate (R5P) in control and fludarabine-treated LCL, CA46, and SUDHL4 cells. (**f**) Bar graphs represent pool size changes in the metabolites with fludarabine treatment in LCL, CA46, and SUDHL4 cells. (**g**) Diagrammatic summary of the metabolic fate of glucose metabolism with fludarabine treatment in the lymphoma cells. The error bars represent the standard deviation from the mean of experimental triplicates. Statistically significant differences from the comparison between control and fludarabine are denoted by * with *p*-values of <0.05 by student *t* Test.

**Figure 7 cancers-16-03606-f007:**
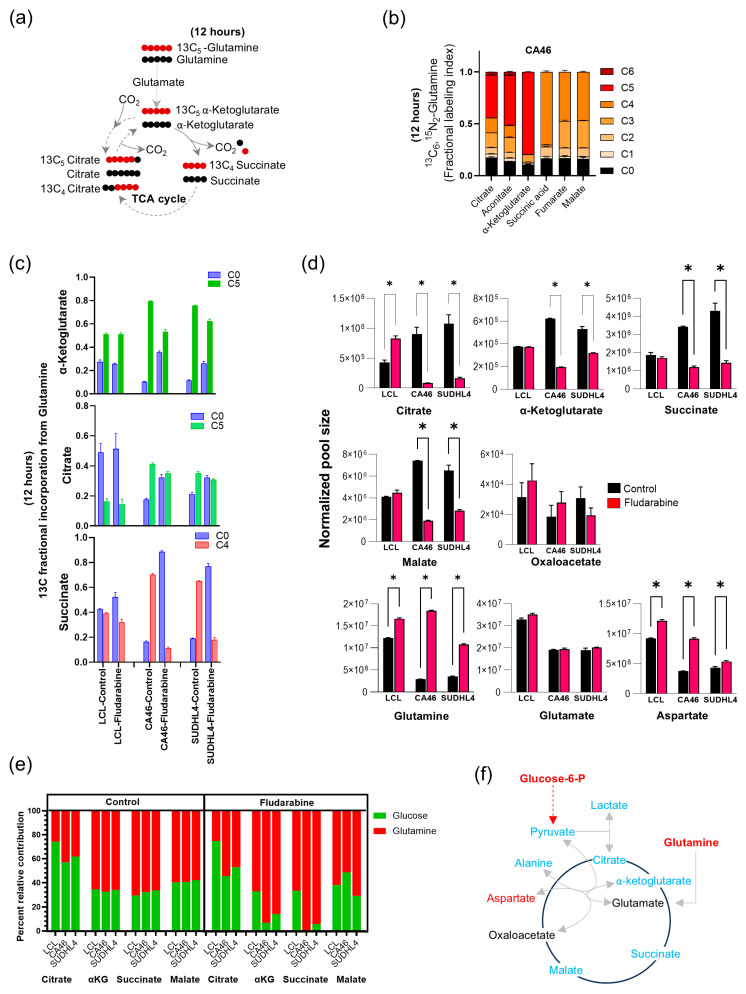
**Fludarabine decreases**^13^C **incorporation from glutamine into TCA cycle in lymphoma.** (**a**) Schematics illustrate patterns of isotopomers expected from ^13^C enriched TCA cycle metabolites derived from ^13^C_5_,^15^N2-Glutamine. (**b**,**c**). The distribution plot represents ^13^C enriched isotopomers representing TCA cycle metabolites in CA46 cells and the bar graphs show ^13^C fractional enrichment changes in TCA cycle intermediates with fludarabine treatment in LCL, CA46 and SUDHL4 cells. (**d**) The bar graphs represent the mean changes in the pool sizes of TCA cycle intermediates and associated metabolites with fludarabine treatment in LCL, CA46 and SUDHL4 cells. (**e**) Relative carbon contribution from glucose and glutamine from ^13^C enriched TCA cycle intermediates, and the effect of fludarabine in LCL, CA46, and SUDHL4 cells are represented as a distribution plot. (**f**) An overview of metabolic pool size changes in lymphoma cells treated with fludarabine is summarized in this illustration based on Figure 6f and Figure 7d, with pool size decreases (in blue) and increases (in red). The error bars in this figure represent the standard deviation from the mean of experimental triplicates. Statistically significant differences from the comparison between control and fludarabine are denoted by * with *p*-values of <0.05 by student *t* Test.

**Figure 8 cancers-16-03606-f008:**
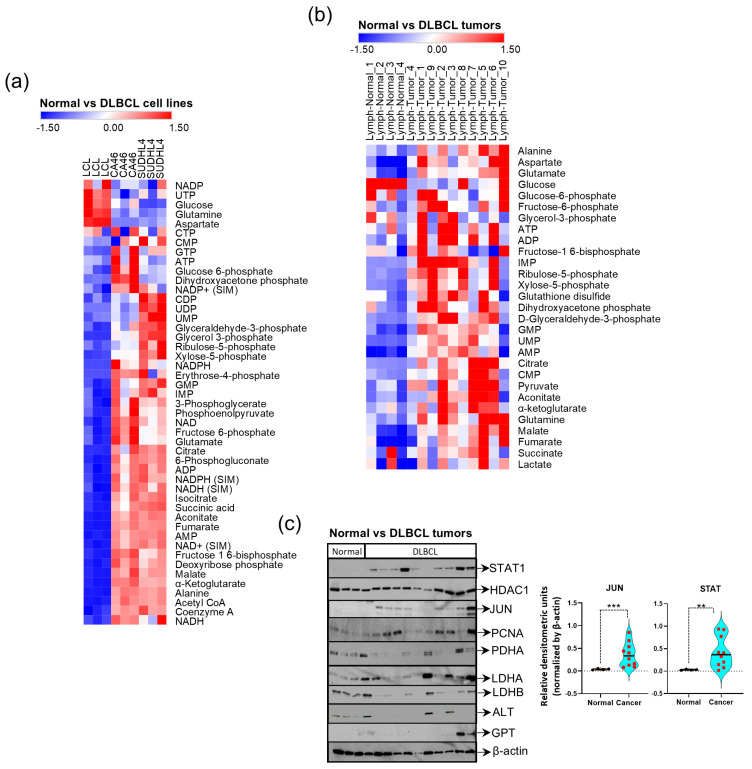
**Warburg paradoxical metabolic features are elevated in lymphoma.** (**a**,**b**). Heatmaps from metabolomic profiling comparing (**a**) primary lymphocyte (LCL) with lymphoma cell lines (CA46 and SUDHL4) and (**b**) normal lymph node and lymphoma tumors show upregulation in the pool sizes of metabolites associated with glucose metabolism, lactate, alanine and nucleotides in lymphoma. Color gradient indicates the absolute mean deviation between pool sizes for each metabolite. (**c**) Western blot analysis comparing lymphoma tumor and normal lymph nodes show that Jun (denoted as ***, *p* < 0.005) and STAT1 (denoted as **, *p* < 0.05) expressions (normalized by β-actin) represented as violin plots, are significantly upregulated in the tumors. Original uncropped blots are presented in Appendix A.

**Figure 9 cancers-16-03606-f009:**
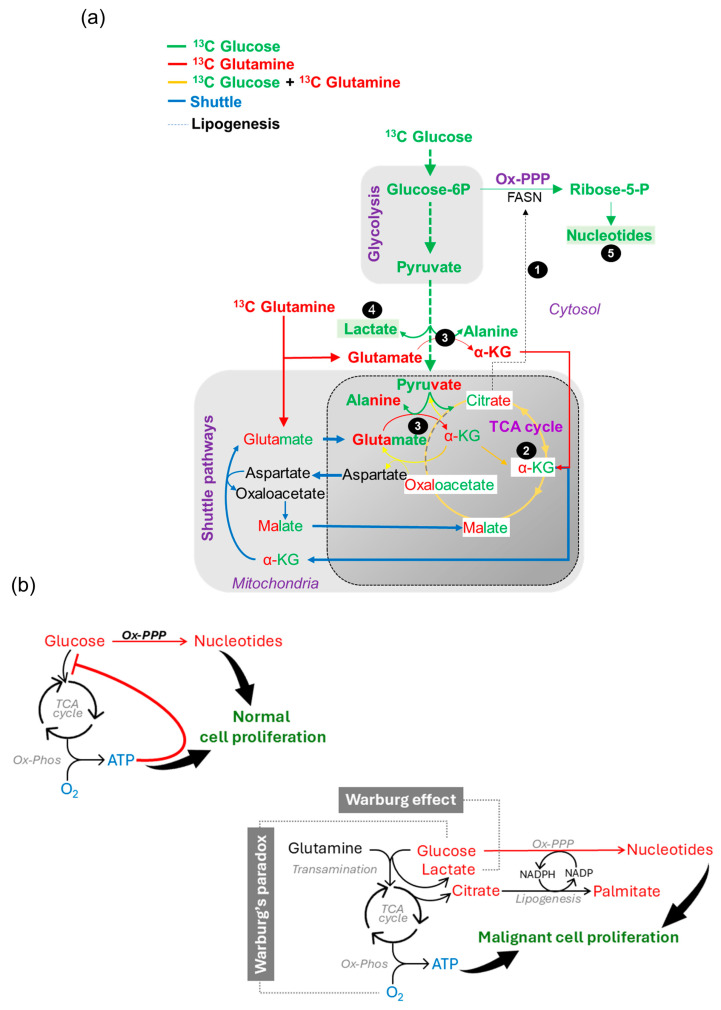
(**a**) Diagrammatic summary illustrating the results from isotopic tracer experiments (highlighting the predicted metabolic sequence of steps, outlined in the results), providing an integrated overview of proliferative metabolic functions in lymphoma. (**b**) Diagrammatic summary of physiological metabolic regulations in proliferation, highlighting the disruption of these regulations by the Warburg phenomenon in malignancy.

## Data Availability

The raw data for transcriptomics analysis performed in this manuscript is available at the NCBI Gene Expression Omnibus database, with the following identifiers: GSE102760 GSE102764, GSE66417, GSE66415, and GSE126768. Metabolomic profiling datasets are included in Appendix A.

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
