# Peer review of "Deciphering the Metabolic Basis and Molecular Circuitry of the Warburg Paradox in Lymphoma"

_cancers, 2024, doi:10.3390/cancers16213606_

Round 1
Reviewer 1 Report
Comments and Suggestions for Authors
First and foremost, the Abstract is too long.
The initial description of Warburg effect in the Summary as well as Abstract does not very accurately describe the phenomenon. Please edit.
The statements ‘Although the literature offers a comprehensive overview of metabolic aspects, it often synthesizes results from experiments on different metabolic steps and various cell types with contradictory features, making it challenging to understand or interpret the biological implications in a cancer-specific context.’ Do not make much sense. – Literature..synthesizes results? Please edit to provide smaller and more clearer statements.
Line 94 – ‘…whether they differ from malignant cells (lymphoma cell lines and tumors) or normal cells’ – do authors mean to say ‘…whether they differ between malignant cells (lymphoma cell lines and tumors) and normal cells’ ?
In the Introduction, authors mention ‘By inhibiting metabolic activity through bioinformatically selected targets, the second approach blocks proliferation selectively in malignant cells, while identifying metabolic functions that become down-regulated as a result.’ This approach makes sense. However, wouldn’t doing the same thing in non-malignant cells better help identify malignancy-specific functions? In this reviewer’s opinion, this represents a flaw in design of experiments / study !
Line 113- did authors perform STR profiling themselves. When was it last performed?
Lines 145-163, authors describe the production of tumor lysates. However, on line 165, they casually mention ‘Whole cell lysates were used in metabolomic profiling experiments..’. How were these cell lysates prepared? My apologies if the information is provided somewhere and I missed it.
Please provide a little more detailed overview of cell proliferation assays.
Line 248 – ‘31 metabolites representing glycolysis, TCA cycle intermediates…’ – do authors mean that they focused on glycoslysis, TCA cycle intermediates and nucleotides for this particular shortlisting? Under a more global and unbiased listing, how prominent these 31 metabolites would be – will some metabolites even feature among top 5 or 10 most up-regulated?
The Results overall are described well.
On line 118, authors mention inhibitor MCT-II while on line 496, they mention MCT-III – which one is correct?
Why are inhibitors CPT1 and MK801 not mentioned in Methods even though they are mentioned in Results (line 496)?
In the methods, authors mention an inhibitor ‘YK-4-274’ but throughout the Results they mention ‘YK-4-279’ – which one is correct? Clearly there is need for better proof-reading!
Author Response
Please see our response to your comment attached.

Reviewer 2 Report
Comments and Suggestions for Authors
This study found that the transamination of pyruvate to alanine supports malignant cell proliferation in lymphoma by maintaining the TCA cycle and reducing glucose oxidation, uncovering a key metabolic mechanism of Warburg’s paradox.However, there are some major concerns that need to be addressed to ensure the findings are convincing.
1.The introduction does not sufficiently present the latest advancements in the field, even though it mentions one of the authors' previous studies.
2.The inhibitors and antibodies for Western blot should include the manufacturers and catalog numbers to ensure the reproducibility of the research findings.
3.The description of the cell cycle staining process in Figure 1 is not very clear. Typically, cells are gated based on FSC and SSC, followed by the removal of doublets, and finally, the cell cycle is determined by fluorescence intensity. The authors need to provide a reason for determining the cell cycle without using PI staining.
4.The MTT assay description is not detailed enough. At the very least, information such as the type of multi-well plate used and the number of cells per well should be included.
5.The authors should provide detailed sample information of the GSE datasets, which is recommended to be presented in a table format.
6.Line 216: "from what" is unclear.
7.In Figure 8, the expression level of the housekeeping gene in the Western Blot results is highly inconsistent, making the statistical analysis of relative expression changes based on this reference gene not very convincing.
8.It is recommended to provide higher high-resolution images.
Author Response
Please see our responses to your comment attached.

Reviewer 3 Report
Comments and Suggestions for Authors
The article entitled “Deciphering the metabolic basis and molecular circuitry of the 2 Warburg’s paradox in lymphoma” by Sathyamurthy et al has systemically elucidated the intersection between glycolysis and TCA cycle coupling in cancer cells through pyruvate transamination. The use of methodology in this research including isotope-labeling-based metabolite profiling within cells of various proliferating phases is rigorous and comprehensive. The work is important in that the biological significance of the Warburg paradox for energy supply and nucleotide biosynthesis is verified by the mechanistic study in the article. I recommend the article to be published after the issue below is addressed:
1. Many abbreviations should be labeled with their full names at their first appearance such as LDH, DLBCL…
2. It's fairly confusing for the readers of the whole metabolic background during the introduction section (became better when I saw figure 9). It would be better to include an illustration scheme (or move part of fig9 to the front) that shows which question needs to be solved for the whole metabolism.
Author Response
Please see our responses to your comments attached.

Round 2
Reviewer 1 Report
Comments and Suggestions for Authors
Thanks for addressing all my concerns.
Reviewer 2 Report
Comments and Suggestions for Authors
All of my concerns have been well addressed.